# Interplay between *TERT* promoter mutations and methylation culminates in chromatin accessibility and *TERT* expression

**Catarina Salgado**[1], **Celine Roelse**[1], **Rogier Nell**[2], **Nelleke Gruis**[1], **Remco van Doorn**[1], **Pieter van der Velden**[2]*

**1** Department of Dermatology, Leiden University Medical Center, Leiden, The Netherlands, **2** Department of Ophthalmology, Leiden University Medical Center, Leiden, The Netherlands

* P.A.van_der_Velden@lumc.nl

**Data Availability Statement:** All relevant data are within the manuscript and its Supporting Information file.

## Abstract

The *telomerase reverse transcriptase* (*TERT*) gene is responsible for telomere maintenance in germline and stem cells, and is re-expressed in 90% of human cancers. CpG methylation in the *TERT* promoter (*TERT*p) was correlated with *TERT* mRNA expression. Furthermore, two hotspot mutations in *TERT*p, dubbed C228T and C250T, have been revealed to facilitate binding of transcription factor ETS/TCF and subsequent *TERT* expression. This study aimed to elucidate the combined contribution of epigenetic (promoter methylation and chromatin accessibility) and genetic (promoter mutations) mechanisms in regulating *TERT* gene expression in healthy skin samples and in melanoma cell lines (n = 61). We unexpectedly observed that the methylation of *TERT*p was as high in a subset of healthy skin cells, mainly keratinocytes, as in cutaneous melanoma cell lines. In spite of the high promoter methylation fraction in wild-type (WT) samples, *TERT* mRNA was only expressed in the melanoma cell lines with either high methylation or intermediate methylation in combination with *TERT* mutations. *TERT*p methylation was positively correlated with chromatin accessibility and *TERT* mRNA expression in 8 melanoma cell lines. Cooperation between epigenetic and genetic mechanisms were best observed in heterozygous mutant cell lines as chromosome accessibility preferentially concerned the mutant allele. Combined, these results suggest a complex model in which *TERT* expression requires either a widely open chromatin state in *TERT*p-WT samples due to high methylation throughout the promoter or a combination of moderate methylation fraction/chromatin accessibility in the presence of the C228T or C250T mutations.

## Introduction

Approximately 90% of all human cancers share a transcriptional alteration: reactivation of the telomerase reverse transcriptase (*TERT*) gene [1, 2]. *TERT* encodes the catalytic subunit of the ribonucleoprotein telomerase and is capable of extending the repetitive, non-coding DNA sequence on terminal ends of chromosomes, the telomeres. As the single-stranded 5' ends of chromosomes are shortened with each cellular division, telomeres prevent loss of coding chromosomal DNA [3–6]. Telomerase is only transcribed in a subset of stem cells in growing or

**Funding:** This project has received funding from the European Union's Horizon 2020 research and innovation programme under the grant agreement No 641458 (http://melgen.org/). RN is supported by the European Union's Horizon 2020 research and innovation program under grant agreement No 667787 (UM Cure 2020 project, https://www.umcure2020.org/en/). The funders had no role in study design, data collection and analysis, decision to publish, or preparation of the manuscript.

**Competing interests:** The authors have declared that no competing interests exist.

renewing tissues, but through reactivation of telomerase expression, cells can extend telomeres or prevent telomeres shrinkage. This is termed telomere maintenance, which is one of the hallmarks of cancer, and allows subsequent indefinite proliferation and immortalization [3, 6–8].

Since the *MYC* oncogene has firstly been identified to activate telomerase, a variety of epigenetic or genetic mechanisms in the gene body or *TERT* promoter (*TERT*p) have followed, such as CpG methylation, histone modifications, mutations, germline genetic variations, structural variations, DNA amplification or chromosomal rearrangements [3, 5, 7].

A widely investigated mechanism that could induce *TERT* reactivation is the presence of mutations in the gene promoter [7, 9]. Horn and Huang *et al.* identified two mutually exclusive *TERT*p point mutations that are correlated to *TERT* mRNA expression by creating binding motifs for the transcription factor E26 transformation-specific/ternary complex factor (ETS/TCF) [7, 9]. These mutations, chr5:1,295,228 C>T and chr5:1,295,250 C>T in hg19 (−124 bp and −146 bp from the translation start site, respectively), henceforth respectively dubbed C228T and C250T, were first identified in melanoma. Furthermore, these mutations showed high prevalence in and were correlated with poor prognosis of cutaneous melanomas [4, 5, 10–12].

An additional mechanism by which a gene can be made accessible to transcription factors, facilitating gene expression, is hypomethylation of promoter CpG islands, a hallmark of euchromatin [13, 14]. Methylation located in the gene body, however, shows a positive correlation with active gene expression [15]. In stark contrast to most genes, *TERT*p hypermethylation may also allow gene expression since transcriptional repressors rely on unmethylated promoter CpGs, such as CCCTC-binding factor (CTCF)/cohesin complex or MAZ [16–18]. As such, in combination with transcription factor binding, dissociation of the repressor may result in *TERT* expression [3, 16, 19, 20]. Castelo-Branco *et al.* proposed that methylation of a specific CpG site in *TERT*p, cg11625005 (position 1,295,737 in hg19) was associated with paediatric brain tumours progression and poor prognosis [20]. This finding was later supported by the study from Barthel *et al.*, in which the CpG methylation was found to be correlated with *TERT* expression in samples lacking somatic *TERT* alterations and to be generally absent in normal samples adjacent to tumour tissue [3].

Chromatin organisation, its plasticity and dynamics at *TERT*p region have been reported as relevant players in regulation of gene expression by influencing the binding of transcription factors [21, 22]. Cancer cells are positively selected to escape the native repressive chromatin environment in order to allow *TERT* transcription [23].

In the present study, we aim to elucidate the interaction of genetic and epigenetic mechanisms in regulation of *TERT*p. We approach this by using novel droplet digital PCR (ddPCR)-based assays [24]. Human-derived benign skin cells (keratinocytes, dermal fibroblasts, melanocytes, skin biopsy samples and naevi) and melanoma cell lines were analysed. The *TERT*p mutational status was assessed along with the absolute presence of methylation in the *TERT*p at a CpG-specific resolution. The effect of chromatin accessibility in *TERT* expression was evaluated in a subset of cultured melanoma cell lines.

## Results

### NGS-based deep bisulfite sequencing and development of a ddPCR assay to assess *TERT*p methylation fraction

We first aimed to quantitatively measure the *TERT*p methylation at a CpG-specific resolution in primary skin samples and melanoma cell lines. DNA of 44 primary skin biopsy samples and melanoma cell lines was bisulfite-converted (BC) and analysed using NGS-based deep bisulfite sequencing to assess the methylation fraction (MF) in a region of *TERT*p encompassing 31

CpG sites. The *TERT*p MF was high in some healthy skin samples, such as normal skin (~30%), naevi (~30%) and cultured keratinocytes (~50%). In the latter group, in fact, the MF was as high as in cutaneous melanoma cell lines (Figs 1 and 7A). In contrast, the fibroblasts and low-passage cultured melanocytes show the lowest MF observed in this cohort. Since the cutaneous melanoma originates from melanocytes of the skin, we found the difference in MF between normal melanocytes and cutaneous melanoma cells quite remarkable.

In order to validate the *TERT*p MF obtained through NGS in a quantitative manner, we have developed a ddPCR assay (Fig 2A) using methylation-sensitive restriction enzymes (MSREs) HgaI and AvaI, which recognise the CpG on position 1,295,737 (cg11625005) and 1,295,731 in hg19, respectively. Castelo-Branco *et al.* showed that methylation of the cg11625005 in *TERT*p, was associated with tumour progression and poor prognosis of childhood brain tumours [20]. Barthel *et al.* affirmed a correlation between methylation and *TERT* expression in samples lacking somatic *TERT* alterations and a lower methylation level in normal samples [3]. Indeed, in our study, the MF of fibroblasts was as low as that of the unmethylated control DNA, whereas that of the keratinocytes was higher than most of the cutaneous melanoma cell lines (Fig 2B). The MF of cg11625005 (position 1,295,737) obtained through NGS and by ddPCR were highly correlated ($R^2$ = 0.82, p<0.001) (Fig 2C). The MF of 1,295,731 assessed through ddPCR even yielded a stronger correlation ($R^2$ = 0.96, p<0.001) (Fig 2D).

## Absence of correlation between methylation fraction and *TERT* expression

Cancer cells are commonly characterised by hypermethylation of promoter CpG islands resulting in repression of tumour suppressor genes. However, in *TERT*, promoter

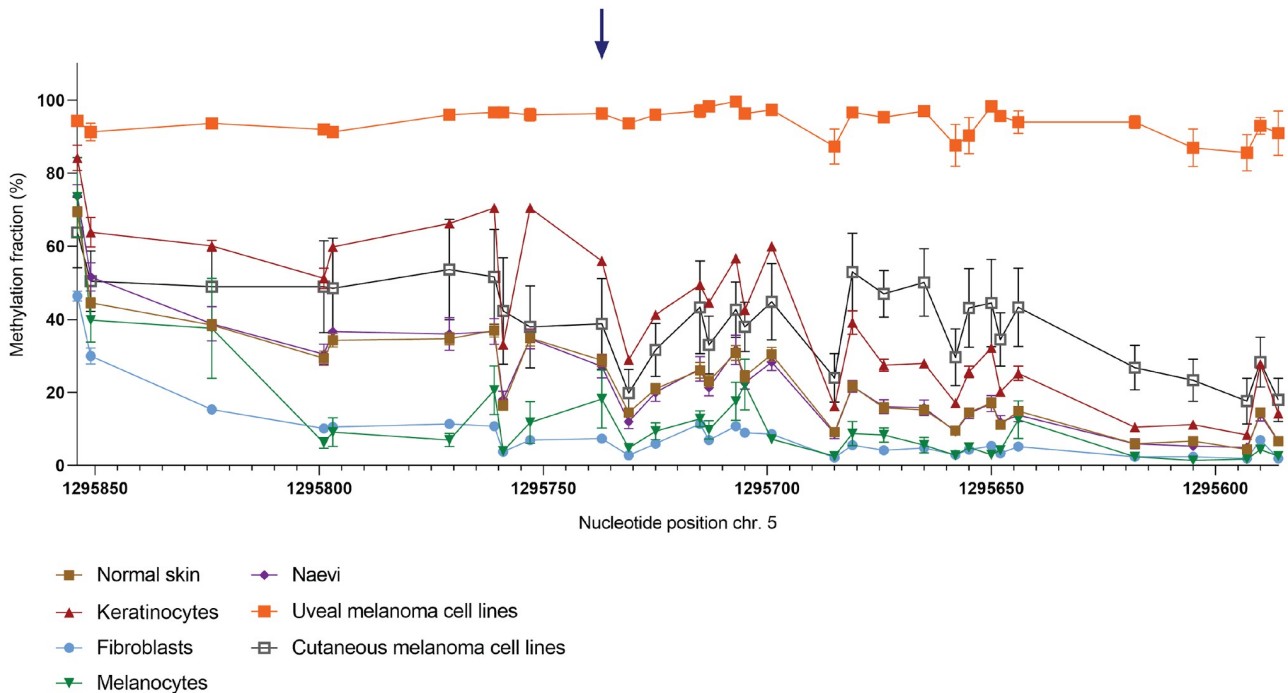

**Fig 1. Methylation fraction (MF) of 31 CpG sites around cg11625005 in 35 primary skin samples and 9 melanoma cell lines.** DNA samples were bisulfite-converted (BC) and analysed through NGS-based deep sequencing. Connected scatter plot representing the MF per cell type group in absolute distance between measured CpG sites. Blue arrow: cg11625005 (position 1,295,737). Samples included: fibroblasts (n = 5), melanocytes (n = 5), naevi (n = 6), normal skin samples (n = 11), keratinocytes (n = 8), cutaneous melanoma cell lines (n = 6) and uveal melanoma cell lines (n = 3).

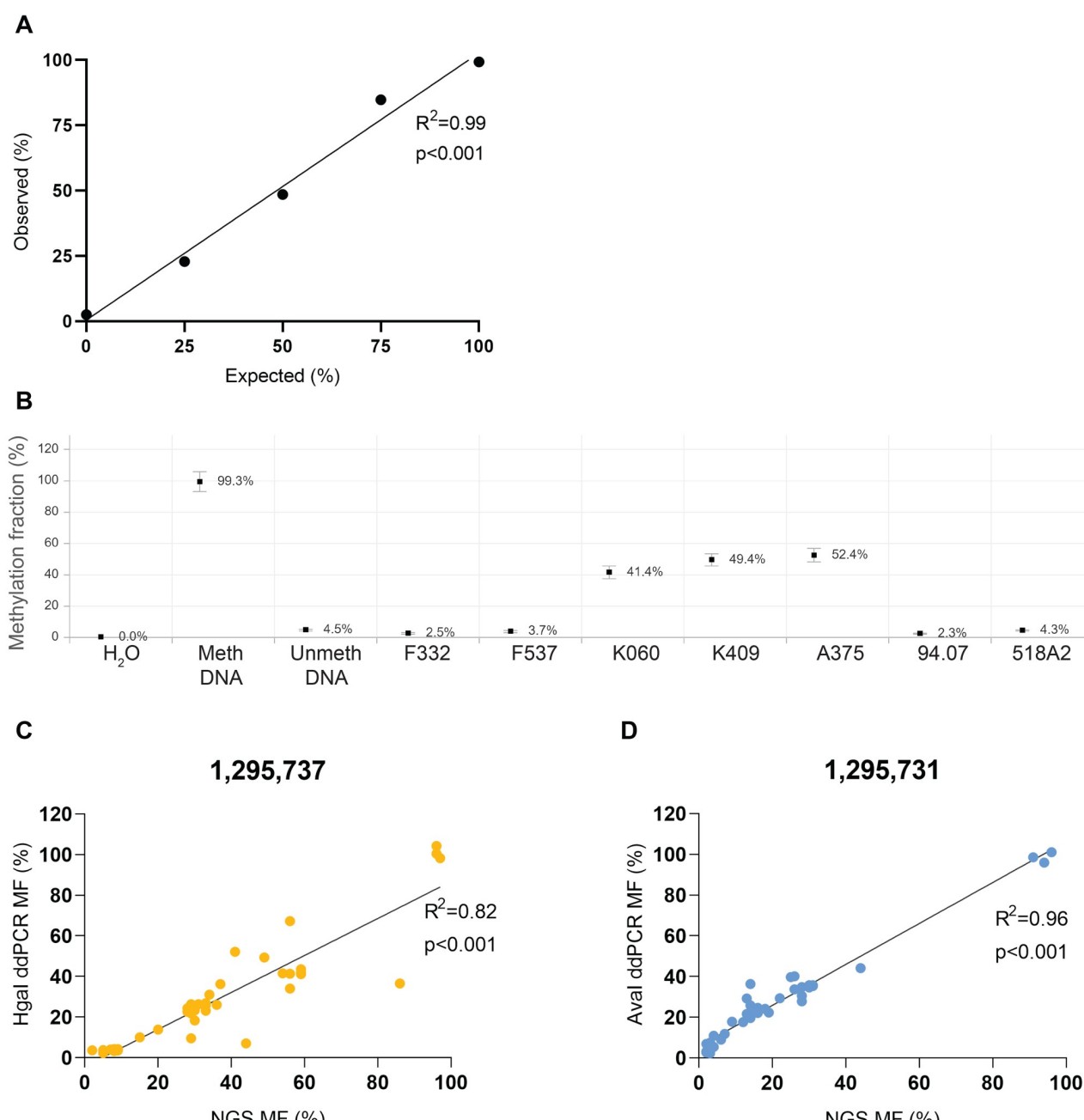

**Fig 2. Methylation fraction (MF) analysed through ddPCR.** MDNA and UDNA are commercially available methylated and unmethylated DNA, respectively. A. Calibration curve using different expected ratios (25%, 50% and 75%) of methylated DNA and F332 to demonstrate the quantitative capacity of ddPCR. Linear regression and correlation analysis were performed to compare the expected to observed ratios ($F_{(1,3)}$ = 209.2, r = 0.99, p<0.001). B. MF of cg11625005 in a subset of healthy primary skin samples–fibroblasts (F332 and F537) and keratinocytes (K060 and K409) and cutaneous melanoma cell lines (A375, 94.07 and 518A2) incubated with MSRE HgaI. MF was plotted with 95% CI through RoodCom WebAnalysis (version 1.9.4). C & D. Correlation plots between MF obtained through golden standard NGS-based deep bisulfite sequencing versus ddPCR using either the MSRE HgaI (C.) or AvaI (D.), which digest unmethylated CpG in position 1,295,737 and 1,295,731, respectively, in a batch of 44 samples: fibroblasts (n = 5), melanocytes (n = 5), naevi (n = 6), normal skin samples (n = 11), keratinocytes (n = 8), cutaneous melanoma cell lines (n = 6) and uveal melanoma cell lines (n = 3). Linear regression and correlation analysis were performed ($F_{(1,40)}$ = 178.1, r = 0.90 and $F_{(1,41)}$ = 934.4, r = 98, respectively, p<0.001).

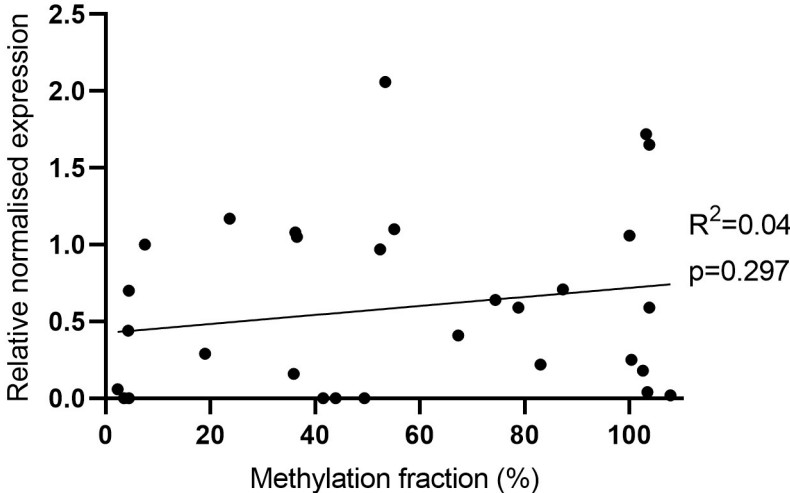

**Fig 3. Correlation between methylation fraction (%) and *TERT* mRNA expression in total of 31 samples: Fibroblasts (n = 3), melanocytes (n = 1), keratinocytes (n = 2), cutaneous melanoma cell lines (n = 19) and uveal melanoma cell lines (n = 6).** Linear regression and correlation analysis were performed ($F_{(1,29)}$ = 1.13, r = 0.19, ns p = 0.297).

hypermethylation was found to be associated with higher expression, since CTCF repressors of *TERT* transcription do not bind methylated sequences [3, 16, 17, 19]. In our sample cohort, there was no correlation between *TERT* methylation of cg11625005 and mRNA expression (n = 31, Figs 3 and 7B).

## Evaluation of *TERT*p mutations in a collection of skin samples and melanoma cell lines

Besides promoter methylation, somatic mutations are also known to be correlated with *TERT*p reactivation. Therefore, we characterised the *TERT*p mutational status of the sample cohort. Sanger sequencing on one naevus, fresh skin and cutaneous melanoma cell lines 518A2, 607B, A375, 94.07 and 93.08 revealed melanoma-associated *TERT* C250T and C228T mutations (Fig 4A). Aiming to use the ddPCR method to evaluate the mutational load of the samples, the *TERT* C250T and C228T mutation assays were validated in three samples of which the mutation was identified in sequencing analysis, 518A2, 607B and A375 (Fig 4B). Following the test runs, the C228T and C250T assays were used on the extended sample cohort (n = 61) (S5 Table and Fig 7C). All *TERT*p-mutated samples were cutaneous melanoma cell lines, however OCM8 and 94.13 cutaneous cell lines tested wild-type. The C250T mutation was not present in combination with the C228T mutation in any sample, confirming that the mutations are mutually exclusive.

## Absence of correlation between mutational status and *TERT* expression

As the presence of mutations in the gene promoter induces *TERT* reactivation, we assessed the correlation between mutational status with *TERT* mRNA expression (n = 31). When WT and mutated samples (either C228T or C250T) were compared, regardless of origin of the tissue, no significant differences for *TERT* mRNA expression were found (Fig 5). Moreover, *TERT* expression was exclusive to the melanoma cell lines, either with or without *TERT*p mutations (Fig 7B).

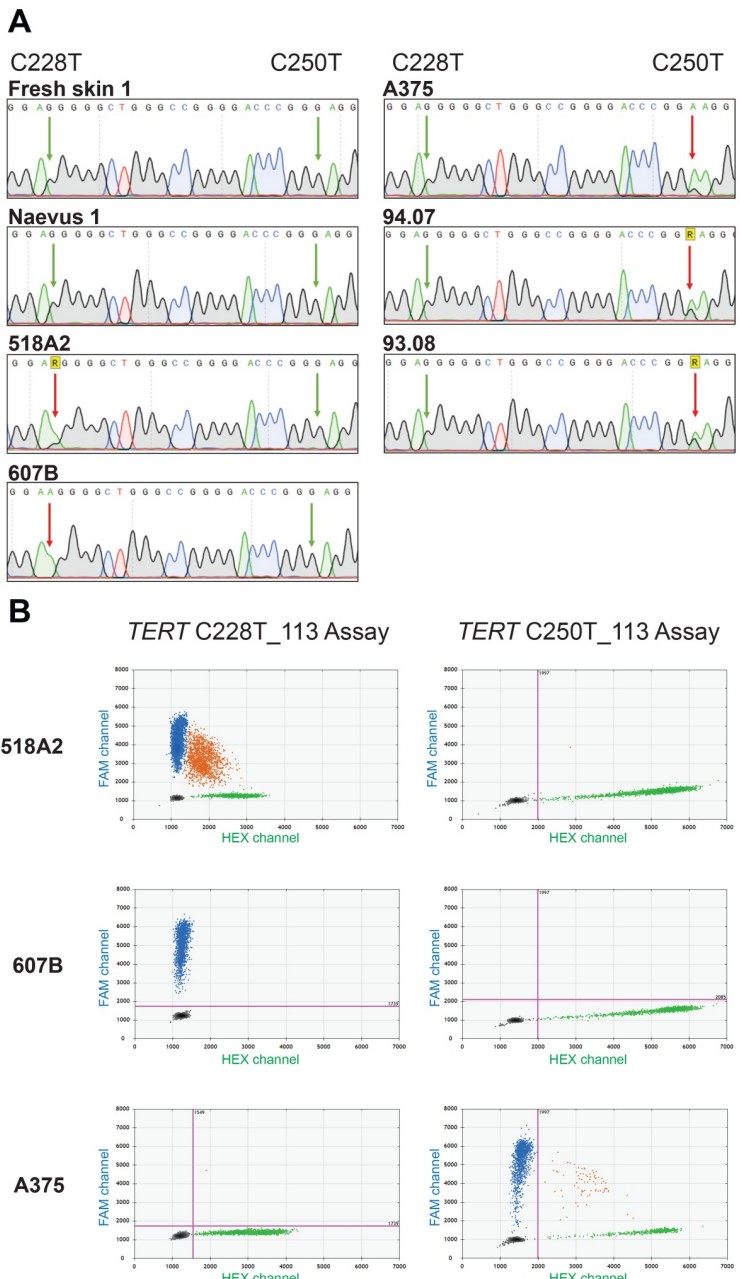

**Fig 4. *TERT*p mutational status of primary skin samples and cutaneous melanoma cell lines.** A. The *TERT*p region encompassing the C228T and C250T mutations was sequenced through Sanger sequencing using McEvoy's [25] *TERT*p forward primer. The *TERT*p region of fresh skin 1, Naevus 1, 518A2, 607B, A375, 94.07, 93.08 is shown. The left and right arrows respectively indicate the positions 1,295,228 and 1,295,250. R: one-letter code for bases G or A; Green arrow: wild-type; red arrow: C>T mutation on the complementary strand. B. Evaluation of *TERT*p mutations through commercial Bio-Rad TERT assays in 518A2, 607B and A375 melanoma cell lines. 2D ddPCR plots of the results from the C228T mutation assay (left) and C250T mutation assay (right). The blue cloud represents mutant copies; the green cloud represents WT copies.

## *TERT* expression is correlated to chromatin accessibility

In contrast to most genes, methylation of the *TERT*p positively correlates with its mRNA expression [3, 16, 17, 19]. Although we were not able to confirm this finding, we investigated

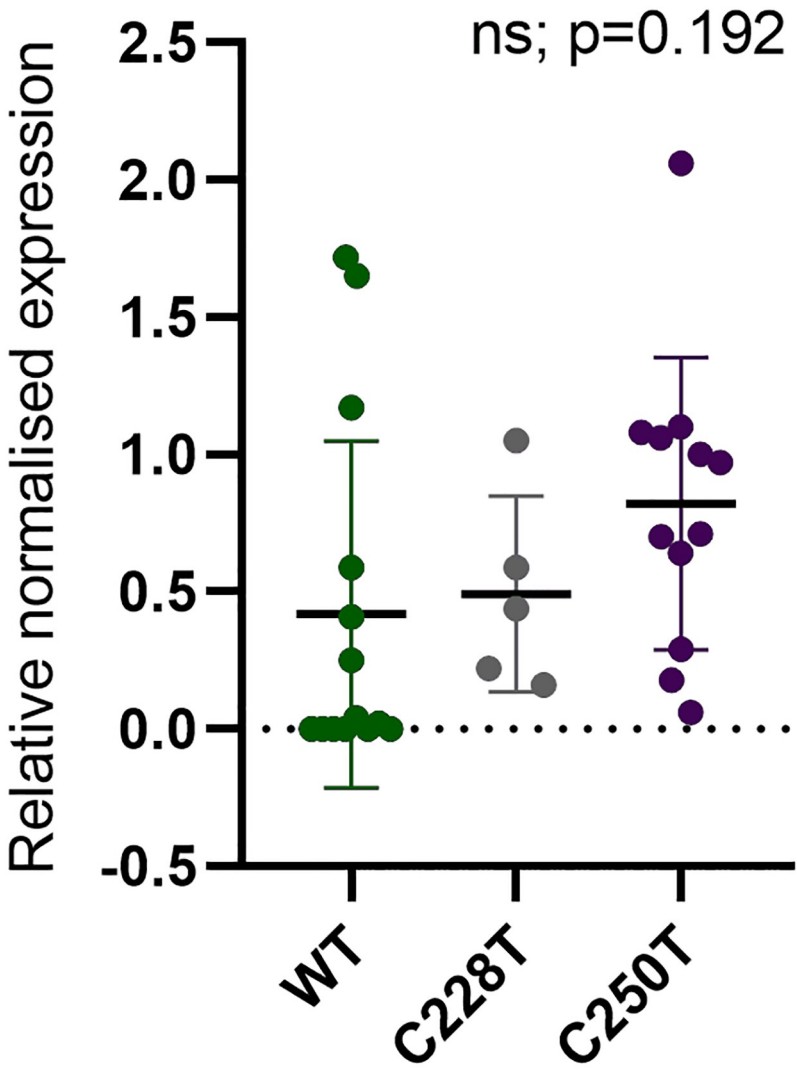

**Fig 5. Correlation between *TERT*p mutational status and *TERT* mRNA expression in total of 31 samples: Fibroblasts (n = 3), melanocytes (n = 1), keratinocytes (n = 2), cutaneous melanoma cell lines (n = 19) and uveal melanoma cell lines (n = 6).** One-way ANOVA ($F_{(2,28)}$ = 1.75, ns p = 0.192).

whether besides promoter methylation, other mechanisms could contribute to chromatin accessibility to transcription factors affecting *TERT*p regulation. Therefore, we analysed chromatin state in a subset of melanoma cell lines (cutaneous, 518A2, 607B, 94.07, A375, 93.08 and OCM8; and uveal, OMM2.5 and Mel270) by ddPCR methodology instead of qPCR for an accurate quantification. The positive control gene *GAPDH*, a housekeeping gene that is generally expressed in all conditions, and thus 100% accessible, was used. The accessibility in the region around cg11625005 shows a high variability, being over 90% in uveal cell lines while being intermediate to low in cutaneous melanoma cell lines (Figs 6A and 7D and S6 Table). When comparing the accessibility around cg11625005 to the methylation fraction of this CpG,

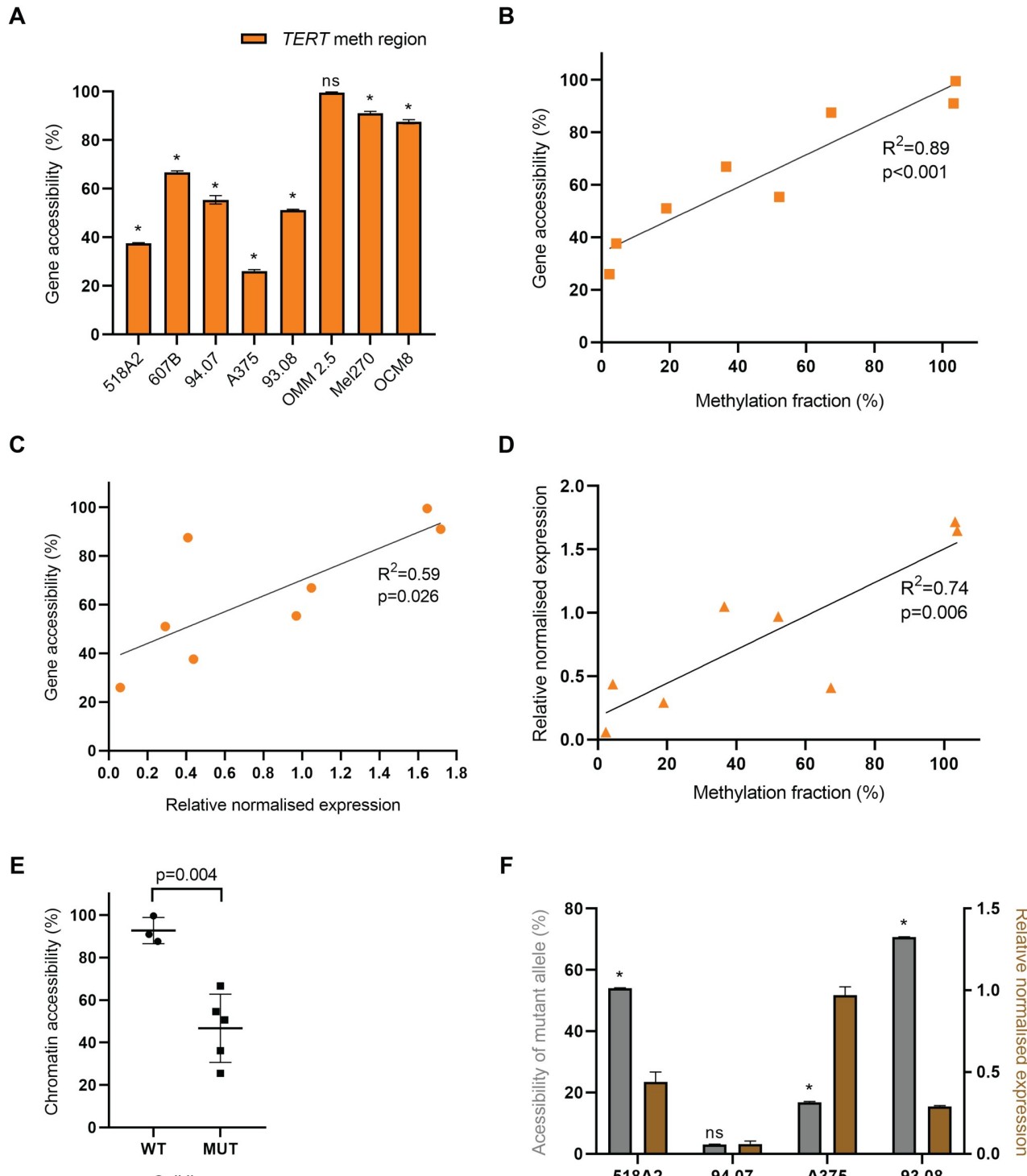

**Fig 6. Accessibility of *TERT*p around cg11625005 in 8 melanoma cell lines.** Cell lines (518A2, 607B, 94.07, A375, 93.08, OMM2.5, Mel270 and OCM8) were analysed with the EpiQ chromatin kit, and ddPCR was performed using primers and probes for positive control gene *GAPDH* and for the *TERT* methylation region, a 231-bp amplicon around cg11625005. Accessibility (%) was calculated by the ratio of the digested sample to its matched undigested sample, subtracted from 1, and subsequently normalised against the positive control *GAPDH*. A. Accessibility of the *TERT* methylation region relative to *GAPDH* (mean ± SD, multiple t-tests, one t-test per cell line, *p<0.001, ns p = 0.149). B & C. Correlation plots of gene accessibility around cg11625005 with the MF (%) of cg11625005 obtained through ddPCR (B), or with normalised expression levels via qPCR (C). Linear regression and correlation analysis were performed ($F_{(1,6)}$ = 49.9, r = 0.95, p<0.001 and $F_{(1,6)}$ = 8.6, r = 0.77, p<0.05, respectively). D. Correlation plot between MF (%) of cg11625005 obtained through ddPCR and normalised expression levels via qPCR. Linear regression and

correlation analysis were performed ($F_{(1,6)}$ = 16.92, r = 0.86, p<0.05). E. Comparison of WT (OMM2.5, Mel270 and OCM8) and mutated (518A2, 607B, 94.07, A375, 93.08) *TERT*-expressing cell lines subsets regarding chromatin accessibility (two-tailed unpaired t-test; t = 4.63, df = 6; p<0.005). F. Accessibility of mutant allele (%) in a subset of 4 *TERT*p-mutated cutaneous cell lines (518A2, 94.07, A375 and 93.08) calculated as described in Material and Methods (mean ± SD, multiple t-tests, one t-test per cell line, ns p = 0.171; *p<0.001) and the *TERT* mRNA expression in the respective cell lines (mean ± SEM).

a significant positive correlation was observed ($R^2$ = 0.89, p<0.001) (Fig 6B). Another positive correlation ($R^2$ = 0.59, p<0.05) was found when comparing the accessibility of the same region to the normalised *TERT* mRNA expression levels in these samples (Fig 6C). In actuality, in this subset of 8 cell lines, the *TERT*p methylation and gene expression show a statistically significant (p<0.05) positive correlation (Fig 6D). The 3 cell lines with higher MF are those with the highest chromatin accessibility (OMM2.5, Mel270 and OCM8). Remarkably, these are also the cell lines with WT-*TERT*p, in which the chromatin accessibility was significantly higher than in the mutated subgroup (Fig 6E).

In addition, we investigated whether the *TERT* accessibility originated from the mutant or the wild-type allele. For this purpose, we assessed the fractional abundance of mutated allele, in the subgroup of 4 *TERT*p-mutated cutaneous cell lines before and after nuclease digestion. 607B cell line was not included since it is homozygous for the mutation and not informative. In 3 out of 4 cell lines preferential digestion of the mutant allele showed that mutated alleles were more accessible than WT alleles (Fig 6F).

## Discussion

By using advanced quantitative methods, we investigated the epigenetic and genetic regulation of *TERT*p in benign and malignant skin cells. Innovative ddPCR-based assays were developed and validated to assess *TERT* promoter methylation and chromatin accessibility. These methods avoid semi-quantitative qPCR and provide absolute quantification even in samples that are challenged by CG-rich DNA sequences, low concentration and integrity.

In the present study the methylation fraction was assessed by NGS interrogating 31 CpGs in the *TERT*p region across 44 healthy, benign and malignant tumour samples. Remarkably, high methylation levels were observed in a variety of normal samples. Mainly in keratinocytes methylation levels exceeded those of cutaneous melanoma cell lines. Previous studies on brain tumours and skin melanoma, observed a general absence of methylation in a specific CpG in *TERT*p, cg11625005, in healthy control samples [3, 20]. Of note, although the authors state absence of methylation we can observe a β-value of ~0.4 (fluorescence ratio provided by Illumina 450K array, ranging from 0 to 1) in their normal samples [3]. In our cohort, the methylation fraction at this CpG was quantified by ddPCR, which validated our results obtained through NGS. Moreover, in our study, methylation of cg11625005 did not stand out across the CpGs in *TERT*p but seemed to be affected along with adjacent CpGs in this genomic region in all samples (Fig 7A). This result suggests that context-related methylation around cg11625005 is biologically relevant as opposed to methylation of one specific CpG.

*TERT*p mutations has been described as a genetic mechanism responsible for induction of *TERT* reactivation [7, 9]. Over the years that followed, a variety of epigenetic or genetic alterations in the gene body or *TERT*p have been identified, such as promoter methylation, mutations, structural variations, DNA amplification, or promoter rearrangements [3, 5, 19].

In accordance with previous studies, regardless of the methylation status, human benign cells neither harbour *TERT*p mutation nor express *TERT*, thereby supporting the principal oncological concept that a benign cell does not undergo undefined proliferation (Fig 8).

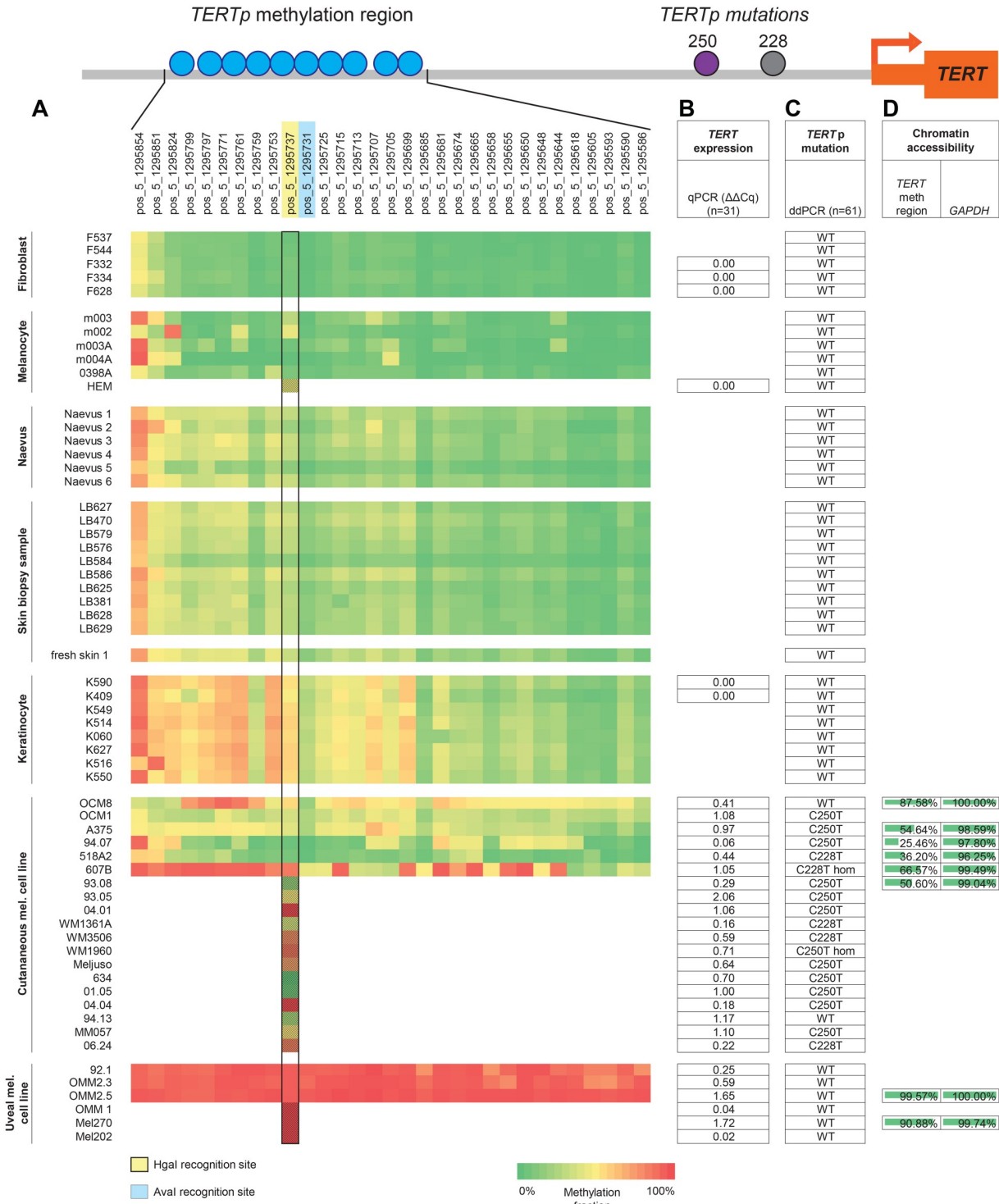

**Fig 7. Results overview.** Schematic representation of *TERT*p with the relative positions of cg11625005 (position 1,295,737 in hg19) to the *TERT*p mutations (position 1,295,228 and 1,295,250) and the transcription start site (TSS). A. Heat-map of methylation fraction (MF) in 31 CpG sites (top) in 44 samples (left). Yellow-marked CpG cg11625005 (position 1,295,737) is recognised by MSRE HgaI. Blue-marked CpG in 1,295,731 is recognised by MSRE AvaI. Black rectangle: MF at the cg11625005 measured either by NGS (clear squares, n = 44) and by ddPCR (patterned squares, n = 17; these samples were not included in the 44-sample batch subjected to NGS). B. *TERT* mRNA expression in 31 samples by qPCR analysed through the ΔΔCT method in Bio-Rad CFX manager software (version 3.1, Bio-Rad). C. *TERT*p mutations evaluated through ddPCR with commercial *TERT* C250T and C228T Mutation Assays in total 61 samples. D. Analysis of the chromatin accessibility in 8 cultured cell lines for *TERT* methylation region using *GAPDH* as a positive control.

Although we have not found a positive correlation between presence of *TERTp* mutations or *TERTp* methylation levels and mRNA expression values, all tumour cell lines showed *TERT* expression, supporting that these mechanisms contribute to telomerase-activation in cancer, separate or in combination [7, 9, 19].

A plethora of histone modifications result in chromatin remodelling that may change accessibility of the *TERT*p to transcription factors, such as ETS/TCF [7]. Schwartz *et al.* state that the degree of chromatin folding is correlated with gene transcription and is thought to impact the regulation of DNA-dependent processes [26]. Therefore, we explored the level of chromatin accessibility and its interaction with methylation levels and mRNA expression in 6 cutaneous and 2 uveal melanoma cell lines. In fact, we found a positive correlation between chromatin accessibility and methylation levels as well as mRNA expression that ultimately explains the correlation between methylation fraction and *TERT* expression. Then, we investigated whether both wild-type and mutant alleles were equally affected by similar patterns of chromatin organization and assessed the mutational fraction upon digestion with nuclease in heterozygous cell lines, assuming that the nuclease only digests DNA open chromatin regions. We could infer that, mutated alleles are more accessible, possibly favouring the binding of transcription factors and consequently *TERT* mono-allelic expression. Our findings in the 518A2 cell line, harbouring the C228T *TERT*p mutation, are similar to the results from a study by Stern *et al.*, in which it was found that the active mutant allele is hypomethylated [27]. These observations are consistent with the canonical influence of methylation on transcriptional regulation. In contrast, 94.07 cell line also presents a very small methylation fraction. However, both alleles were equally resistant to nuclease digestion, which might explain the lowest *TERT* expression levels among all cell lines. Therefore, it still supports the link between local chromatin accessibility and gene regulation [26]. To fully disclose the molecular mechanisms behind *TERT* expression the heterozygous mutant cell lines A375 and 93.08 provide good models as they allow to study a repressed and expressed allele within the same cell.

Another remarkable observation in our study is that in WT *TERT*-expressing uveal melanoma cell lines, the methylation of the whole *TERT*p region is close to 100% with a significantly higher chromatin accessibility compared to *TERT*p-mutated cell lines. Accordingly, Stern *et al.* also demonstrate that cell lines with WT *TERT*p display much higher levels of methylation [27]. These characteristics of WT *TERT*p cell lines may lead to biallelically *TERT* activation under distinct epigenetic conditions from those in mutated *TERT*p.

Interestingly, these results suggest a complex model in which *TERT* expression requires either a widely open chromatin state in *TERT*p-WT samples due to hypermethylation throughout the promoter or mono-allelic expression of the accessible mutated allele in combination with moderate (probably allele-specific) methylation fraction (Fig 8). Furthermore, Huang and colleagues reported that some cancer cell lines show mono-allelic expression of *TERT* even in the absence of *TERT*p mutations [28].

Previous studies have reported the association between *TERT*p hypermethylation and poor patient survival in melanoma and other cancers, indicating that it might be a relevant prognostic marker [20, 27, 29–31]. In primary melanoma it needs to be assessed if *TERT*p methylation is predictive of worse prognosis. Thus, the quantification of *TERT* methylation through ddPCR might be relevant in the clinic to assess patient prognosis.

The dynamics of epigenetic mechanisms in *TERT* genetic regulation is complex. Further investigations are needed to address the correlation of allele-specific differences in chromatin accessibility and promoter methylation with allele-specific mRNA expression.

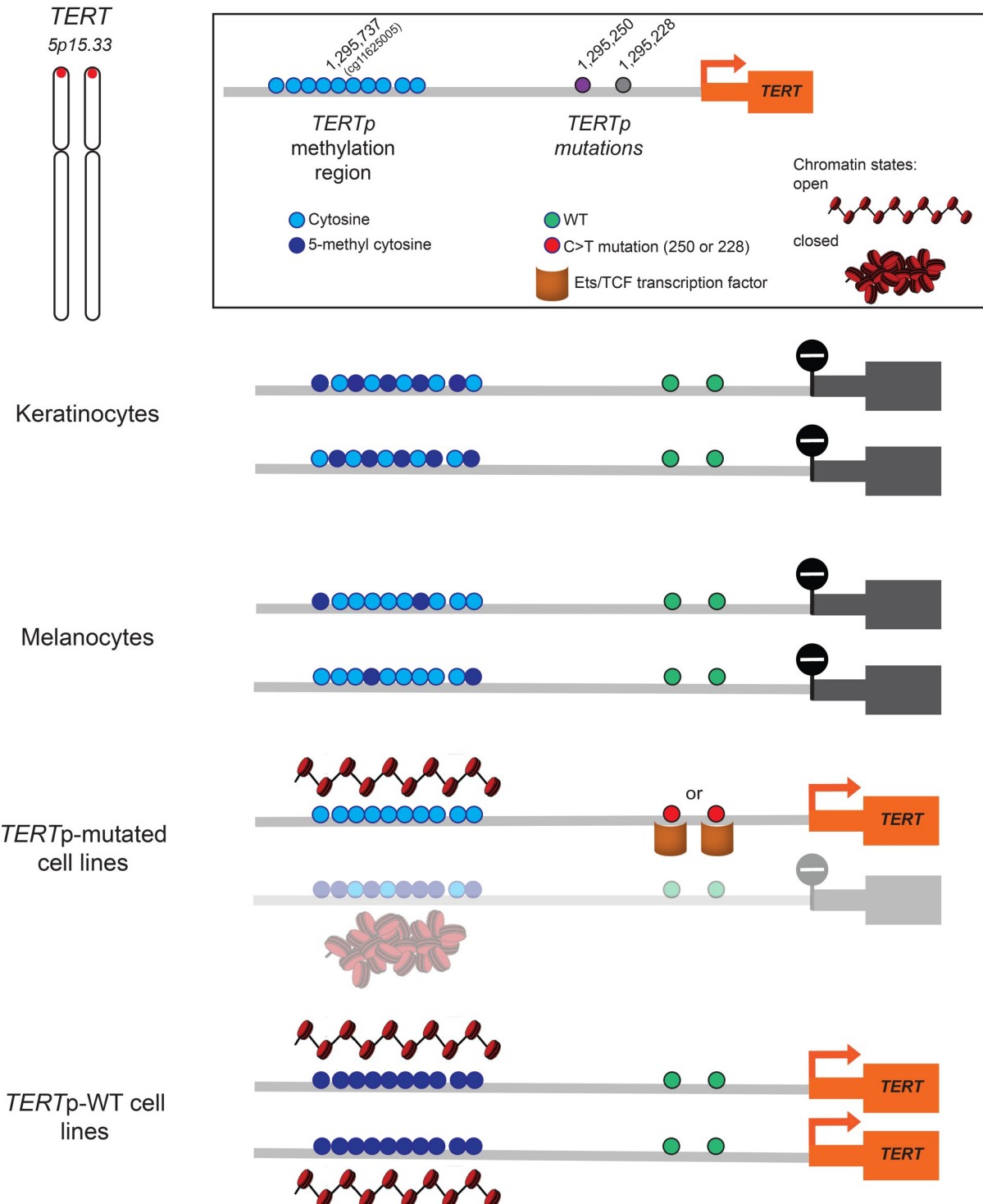

**Fig 8. Proposed model of *TERT* transcriptional regulation.** Regardless of MF at the *TERT*p methylation region, both keratinocytes and melanocytes do not show TERT expression. In *TERT*p-mutated cell lines, an intermediate MF positively correlated with chromatin accessibility, in combination with C228T/C250T *TERT* mutations allows monoallelic *TERT* expression. In *TERT*p-WT cell lines, the MF is close to 100% with a significantly higher chromatin accessibility leading to high expression levels. Chromatin schemes adapted from Schwartz *et al.* [26].

## Material and methods

### Samples, DNA extraction and PCR

Surplus female breast skin and nevi tissues were obtained from 11 and 6 anonymous patients that underwent cosmetic surgery, respectively. Surgeries for mama reduction (performed between 2010 and 2018) and naevi (performed between 2008 and 2009), were conducted according to declaration of Helsinki principles. Epidermis and dermis were separated after removal of adipose tissue followed by enzymatic digestion and primary fibroblast (n = 5) and keratinocyte (n = 8) cell suspensions were obtained and cultured as described before [32]. Keratinocytes were used at passage 2, while fibroblasts were used at passage 3–5.

Low-passage cultured melanocytes (n = 5)–m003, m003A, m002, m004A and 0398A –were cultured as previously described [33]. HEMs were cultured more recently in the medium 254 supplemented with HMGS-2 (Gibco/ThermoFisher) and Penicillin (100 U/ml), and Streptomycin (100 μg/ml; both from Lonza, Verviers, Belgium).

We also included 19 early-passage cutaneous melanoma cell lines derived from metastatic lesions cultured for research purposes and adoptive T-cell transfer [34]. Cell lines were cultured and DNA and RNA extracted between 2017 and 2019. The 518A2, 607B, 04.01, 04.04, 94.13, 93.05, 94.07, 93.08, 634, 01.05, and 06.24 cell lines were a kind gift from Dr. Els Verdegaal (Department of Medical Oncology, LUMC). Meljuso was obtained from Prof. Neefjes (Department of Cell and Chemical Biology, LUMC). WM1361A, WM3506, WM1960 cell lines were a kind gift from Dr. KL Scott (Baylor College of Medicine, Houston, USA). MM057 and A375 were kindly provided by Prof. JC Marine (VIB, Leuven, Belgium). OCM8 and OCM1 were provided by Dr. Mieke Versluis (Department of Ophthalmology, LUMC) [35]. All cell lines were cultured with Dulbecco's modified eagle medium (DMEM, low glucose, pyruvate; Gibco/ThermoFisher) supplemented with 10% FCS, Penicillin (100 U/ml), and Streptomycin (100 μg/ml; both from Lonza, Verviers, Belgium) and glutamax (100X, Gibco).

For the 6 uveal cell lines provided by Dr. Mieke Versluis (Department of Ophthalmology, LUMC), the establishment and culturing conditions have been described before: OMM 1 [36], OMM 2.3, OMM 2.5 and Mel270 [37], Mel202 [38], 92.1 [39]. All cell lines used in our study were tested negative for mycoplasm and recently subjected to STR profiling.

The batch thus consisted of 36 primary skin type samples and 25 melanoma cell lines, totalling 61 samples (Table 1).

DNA was isolated using the QIAamp DNA Blood Mini Kit and the DNeasy Blood & Tissue Kit (both from Qiagen, Hilden, Germany).

Conventional PCR was performed using the PCR-sequencing kit (Thermo Fisher Scientific, Waltham, MA, USA), containing 10X reaction buffer, MgCl2 (50mM), dNTP mix (10nM, Fermentas/Thermo Fisher Scientific), primer mix (900nM each), PlatinumX Taq enzyme (2.5U), 50ng DNA and Aqua B. Braun RNase-free water. A PCR for CG-rich sequences was performed on 50ng DNA using the PCRX Enhancer System (Thermo Fisher Scientific), containing 10X PCRX amplification buffer, MgSO4 (50mM), dNTP mix (10nM), primer mix (900nM each), PlatinumX Taq enzyme (2.5U) and Aqua B. Braun RNase-free water. The samples were amplified in C1000 Touch Thermal Cycler (Bio-Rad Laboratories, Inc., Hercules, CA, USA).

### Promoter methylation determination

**Bisulfite conversion and next-generation sequencing (NGS)-based deep bisulfite sequencing.** In this experiment 44 samples were included: fibroblasts (n = 5), melanocytes (n = 5), naevi (n = 6), normal skin samples (n = 11), keratinocytes (n = 8), cutaneous melanoma cell lines (n = 6) and uveal melanoma cell lines (n = 3). DNA was bisulfite-converted

**Table 1. Samples overview.**

| Control samples | | | | | Melanoma cell lines | |
|---|---|---|---|---|---|---|
| Skin biopsy samples | Fibroblasts | Melanocytes | Keratinocytes | Naevi | Cutaneous | Uveal |
| LB627 | F537 | m003 | K590 | Naevus 1 | 04.01 | OMM 2.3 |
| LB470 | F544 | m002 | K409 | Naevus 2 | WM1361A | OMM 1 |
| LB579 | F332 | m003A | K549 | Naevus 3 | 93.05 | OMM 2.5 |
| LB576 | F334 | m004A | K514 | Naevus 4 | WM3506 | Mel270 |
| LB584 | F628 | 0398A | K060 | Naevus 5 | WM1960 | Mel202 |
| LB586 | | HEM | K627 | Naevus 6 | Meljuso | 92.1 |
| LB625 | | | K516 | | 634 | |
| LB381 | | | K550 | | OCM8 | |
| LB628 | | | | | OCM1 | |
| LB629 | | | | | 518A2 | |
| Fresh skin 1 | | | | | 607B | |
| | | | | | 94.07 | |
| | | | | | A375 | |
| | | | | | 93.08 | |
| | | | | | 94.13 | |
| | | | | | 01.05 | |
| | | | | | 04.04 | |
| | | | | | MM057 | |
| | | | | | 06.24 | |

(BC) using the EZ DNA Methylation™ Kit (Zymo Research, Irvine, CA, USA) according to the manufacturer protocol (version 1.2.2). BC samples were amplified using the PCRX Enhancer System in the program: 1 cycle of 95˚C for 3 minutes, 8 cycles of 95˚C for 30 seconds, 58˚C for 30 seconds, reducing 1˚C/cycle, and 68˚C for 1 minute, then 36 cycles of 95˚C and 53˚C for 30 seconds each, and 68˚C for 1 minute, followed by 1 cycle of 68˚C for 3 minutes. Tailed primers were used for amplification (900nM each; S1 Table). Samples were sequenced through next-generation sequencing (NGS), MiSeq, 2x300bp paired-end, at Leiden Genome Technology Centre (LGTC). Bisulfite sequencing reads were quality trimmed using PRINSEQ (v0.20.4 lite) and aligned to GRCh37 using Bismark (v0.20.0) and Bowtie 2 (v2.3.4.3) [40–42].

**Novel design of a ddPCR assay using methylation-sensitive restriction enzymes (MSREs) to determine *TERT*p methylation fraction.** The methylation fraction (MF) of the CpG (cg11625005) in position 1,295,737 was determined by an in-house designed ddPCR assay in combination with HgaI methylation-sensitive restriction enzyme (MSRE) that cleaves this CpG when unmethylated, as described by Nell *et al.* [24]. 100ng DNA sample was incubated with HgaI (2U/μl) and appurtenant 10X NEBuffer 1.1 (both from New England Biolabs, Bioké, Leiden, The Netherlands) for 60 minutes at 37˚C and 65˚C for 20 minutes. To assess the MF of a CpG adjacent to cg11625005, located in 1,295,731, the MSRE AvaI (10U/μl; New England Biolabs) was employed, which recognises this CpG and cleaves it when unmethylated. Incubation of the DNA samples with AvaI was performed with 10X CutSmart buffer for 15 minutes at 37˚C and subsequently 65˚C for 20 minutes. For ddPCR reaction, 60ng DNA digested or undigested by HgaI, 2x ddPCR SuperMix for Probes (no dUTP), primers (900nM each), a FAM-labelled in-house-designed probe for the CpG site of interest (250nM, Sigma, St. Louis, MO, USA), and 20X HEX-labelled CNV *TERT* reference primer/probe (Bio-Rad) for total *TERT* amplicon count. The primers and probe sequences are presented in S2 Table. The amplification protocol used: 1 cycle of 95˚C for 10 minutes, 40 cycles of 94˚C for 30

seconds and 60˚C for 1 minutes, and 1 cycle of 98˚C for 10 minutes, all at ramp rate 2˚C/s. Droplets were analysed through a QX200 droplet reader (Bio-Rad) using QuantaSoft software version 1.7.4 (Bio-Rad). Raw data was uploaded in online digital PCR management and analysis application Roodcom WebAnalysis (version 1.9.4, https://www.roodcom.nl/webanalysis/) [24], in which the MF was calculated by dividing the CNV of the digested sample with that of the paired undigested sample.

## Assessment of mutational status

**Sanger sequencing.** The presence of the C228T and C250T *TERT*p mutations in some samples was evaluated by conventional Sanger sequencing. DNA samples were amplified through the PCRX Enhancer System (Thermo Fisher Scientific) using primers (Sigma-Aldrich) and amplification program described by McEvoy *et al.* [25].

**Mutation analysis using commercial TERT C250T and C228T mutation assays.** For most of the samples, the *TERT*p mutations were detected by the ddPCR technique according to protocol described by Corless *et al.* [43], using the *TERT* C250T_113 Assay and C228T_113 Assay (unique assay ID dHsaEXD46675715 and dHsaEXD72405942, respectively; Bio-Rad). Both assays include FAM-labelled probes for the C250T and C228T mutations respectively, HEX-labelled wild-type (WT) probes, and primers for a 113-bp amplicon that encompasses the mutational sites. The ddPCR reaction mix comprised 1X ddPCR Supermix for Probes (No dUTP), Betaine (0.5M; 5M stock), EDTA (80mM; 0.5M stock, pH 8.0, Thermo Fisher Scientific), CviQI restriction enzyme (RE; 2.5U; 10U/μl stock, New England BioLabs), the *TERT* assay, and 50ng DNA. Droplets were generated in QX200 AutoDG system (Bio-Rad) and amplified in T100 Thermal Cycler (Bio-Rad) according to the recommended cycling conditions and analysed through a QX200 droplet reader (Bio-Rad) using QuantaSoft software version 1.7.4 (Bio-Rad).

## Chromatin accessibility

**Cell culture and treatment to assess chromatin states.** Cutaneous melanoma cell lines A375, 518A2, 607B, 94.07, 93.08, OMM2.5, Mel270 and OCM8 were cultured for 22 days in 9-cm Cellstar® cell culture dishes (Greiner Bio-One GmbH, Frickenhausen, Germany) with Dulbecco's modified eagle medium (DMEM; Sigma-Aldrich) supplemented with 10% FCS, Penicillin (100U/ml), and Streptomycin (100μg/ml; both from Lonza, Verviers, Belgium) until roughly 95% confluent. Then, different densities (10,000, 20,000, 40,000 and 80,000 cells) of the above-mentioned cell lines were seeded in duplicate into a 48-well plate (Corning Costar, Sigma-Aldrich) required for the EpiQ chromatin assay. The EpiQ™ Chromatin Analysis Kit (Bio-Rad) was performed according to manufacturer's instructions. Briefly, after 2 days each cell line was 85%-95% confluent. The cells were permeabilised and treated with EpiQ chromatin digestion buffer with or without nuclease for 1 hour at 37˚C. Following incubation with EpiQ stop buffer for 10 minutes at 37˚C, the DNA samples were purified using alcohol and DNA low- and high-stringency wash solutions. The genomic DNA was eluted in DNA elution solution.

**Novel design of a ddPCR assay to assess chromatin opening state.** The analysis was performed using ddPCR rather than qPCR, to achieve quantifiable results using *GAPDH* expression as positive control. The reaction mix consisted of 2x ddPCR Supermix for Probes (No dUTP, Bio-Rad), 20x HEX-labelled CNV *TERT* reference primer/probe (Bio-Rad), 50ng DNA, and primers (900nM each) and FAM-labelled probes (250nM) for *GAPDH*, or the methylation region around cg11625005 (S3 Table). Samples were amplified according to the program of the CNV *TERT* reference primer/probe as described. Gene accessibility was

quantified by the digestion fraction between the digested and undigested samples, subtracted from 1, multiplied by 100.

**Allele-specific chromatin accessibility.** The mutational fraction upon digestion with nuclease (EpiQ™ Chromatin Analysis Kit aforementioned) was assessed in cutaneous melanoma cell lines with heterozygous *TERT*p mutations, 518A2, 94.07, A375 and 93.08. The analysis was performed by ddPCR using the *TERT* C250T_113 Assay and C228T_113 Assay (unique assay ID dHsaEXD46675715 and dHsaEXD72405942, respectively; Bio-Rad) as described above. The mutation fraction from undigested and digested samples were compared and the accessibility of mutant allele was calculated as follows:

$$\text{Accessibility of mutant allele}$$
$$= \frac{\text{mutational fraction undigested} - \text{mutational fraction digested}}{\text{mutational fraction undigested}} \times 100$$

## RNA isolation, cDNA synthesis and quantitative real-time PCR

RNA was obtained using the FavorPrep Tissue Total RNA Extraction Mini Kit (Favorgen Biotech, Vienna, Austria) according to manufacturer's instructions for animal cells. cDNA was synthesised through the iScript™ cDNA Synthesis Kit (Bio-Rad) according to recommended protocol. *TERT* mRNA expression was assessed by qPCR performed with 3.5ng DNA, IQ SYBR Green Supermix (2x; Bio-Rad), and 0.5μM PCR primers (Sigma-Aldrich; S4 Table) in a Real-Time PCR Detection System CFX96 (Bio-Rad) and normalised to reference gene expression (*RPS11*, *TBP* and *CPSF6*, S4 Table). Data was analysed through the ΔΔCT method in Bio-Rad CFX manager software (version 3.1, Bio-Rad).

## Statistical analysis

In this study we used the GraphPad Prism software (version 8.0.1 for Windows, GraphPad Software, CA, USA) to perform all the statistical tests. Prism 8 has a wide library of analysis and in our paper we have used the linear regressions and correlations (in Figs 2A, 2C, 2D, 3; 6B, 6C, 6D), one-way ANOVA (Fig 5) and multiple t-tests without correction for multiple comparisons, one t-test per cell line, *$p < 0.001$). (Fig 6A and 6F) and two-tailed unpaired t-test (Fig 6E). A p-value$< 0.05$ was considered statistically significant.

The methylation fraction obtained using ddPCR was calculated with 95% confidence interval by dividing the CNV of the digested sample with that of the paired undigested sample. Raw data was uploaded in online digital PCR management and analysis application Roodcom WebAnalysis (version 1.9.4, https://www.roodcom.nl/webanalysis/) [24] (in Fig 2B).

## Ethics statement

The study was conducted according to the Declaration of Helsinki Principles.

Naevi samples from 6 patients (excised between 2008 and 2009) were accessed anonymously from the biobank of the Department of Dermatology, LUMC and was approved by the Leiden University Medical Center institutional ethical committee (05–036).

Surplus female breast skin was obtained from 11 anonymous patients that underwent surgeries for mama reduction (performed between 2010 and 2018). This entails patient consent was not required since the surplus tissue was considered as waste material. Experiments were conducted in accordance with article 7:467 of the Dutch Law on Medical Treatment Agreement and the Code for proper Use of Human Tissue of the Dutch Federation of Biomedical

Scientific Societies (https://www.federa.org/codes-conduct). As of this national legislation, coded tissue samples can be used for scientific research purposes when no written objection is made by the informed donor. Therefore, additional approval of an ethics committee regarding scientific use of surplus tissue was not required.

## Supporting information

**S1 Table. Tailed primers used for amplification of 325-bp region in bisulfite-converted samples.**
(XLSX)

**S2 Table. Primers and probe sequences to amplify the 106-bp amplicon in a novel design of a ddPCR assay to determine the methylation fraction.**
(XLSX)

**S3 Table. Primers and probe sequences to amplify the 231-bp region encompassing 31 CpG sites around the cg11625005 in a novel ddPCR assay to assess the chromatin state.**
(XLSX)

**S4 Table. Primer and probe sequences for *TERT* expression in qPCR.**
(XLSX)

**S5 Table. Overview of the methylation fraction (measured by ddPCR and NGS), mutational status and *TERT* mRNA expression of our sample cohort (n = 61).**
(XLSX)

**S6 Table. Overview of the methylation fraction (measured by ddPCR and NGS), mutational status and *TERT* mRNA expression and chromatin accessibility in the subset of melanoma cell lines present of our cohort (n = 25).**
(XLSX)

**S7 Table. Raw data used in the Fig 5.**
(XLSX)

**S8 Table. Raw data used in Fig 6.**
(XLSX)

**S9 Table. Results overview.**
(XLSX)

## Acknowledgments

We thank Wim Zoutman, Abdoel el Ghalbzouri, AG Jochemsen and Mijke Visser for useful discussions, Marion Rietveld, Coby Out and Tim van Groningen for the assistance with cell culturing. We would like to thank Dr. Mieke Versluis (Department of Ophthalmology, LUMC), Dr. Els Verdegaal (Department of Medical Oncology, LUMC), Prof. Neefjes (Department of Cell and Chemical Biology, LUMC), Dr. KL Scott (Baylor College of Medicine, Houston, USA) and Prof. JC Marine (VIB, Leuven, Belgium) for kindly providing melanoma cell lines.

## Author Contributions

**Conceptualization:** Catarina Salgado, Remco van Doorn, Pieter van der Velden.

**Data curation:** Catarina Salgado, Celine Roelse.

**Formal analysis:** Catarina Salgado, Celine Roelse, Rogier Nell.

**Funding acquisition:** Nelleke Gruis, Remco van Doorn, Pieter van der Velden.

**Investigation:** Catarina Salgado.

**Methodology:** Catarina Salgado, Rogier Nell.

**Software:** Rogier Nell.

**Supervision:** Nelleke Gruis, Remco van Doorn, Pieter van der Velden.

**Writing – original draft:** Catarina Salgado, Celine Roelse.

**Writing – review & editing:** Rogier Nell, Nelleke Gruis, Remco van Doorn, Pieter van der Velden.

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
