## [Decision Letter · Decision Letter 0]

31 Dec 2019

PONE-D-19-33667

Interplay between TERT promoter mutations and methylation culminates in chromatin accessibility and TERT expression

PLOS ONE

Dear Ms Salgado,

Thank you for submitting your manuscript to PLOS ONE. After careful consideration, we feel that it has merit but does not fully meet PLOS ONE’s publication criteria as it currently stands. Therefore, we invite you to submit a revised version of the manuscript that addresses the points raised during the review process.

We would appreciate receiving your revised manuscript by Feb 14 2020 11:59PM. To enhance the reproducibility of your results, we recommend that if applicable you deposit your laboratory protocols in protocols.io, where a protocol can be assigned its own identifier (DOI) such that it can be cited independently in the future. For instructions see: http://journals.plos.org/plosone/s/submission-guidelines#loc-laboratory-protocols

We look forward to receiving your revised manuscript.

Kind regards,

Srinivas Saladi, Ph.D.

Academic Editor

PLOS ONE

Additional Editor Comments:

Thank you for submitting the manuscript to Plos One. The authors should address the comments raised by reviewer 2 and the minor comments from reviewer 1, for the manuscript to be considered.

Journal Requirements:

2. Please provide additional information about the 39 primary skin type cell samples and 25 melanoma cell lines used in this work, including the source of the cell lines, history, culture conditions and any quality control testing procedures (authentication, characterisation, and mycoplasma testing). For more information, please see http://journals.plos.org/plosone/s/submission-guidelines#loc-cell-lines.

3. In the ethics statement in the manuscript and in the online submission form, please provide additional information about the 11 normal skin samples and 6 frozen naevi used in your retrospective study, including: a) the source of the cells analyzed in this work (e.g. hospital, institution or public biobank) and b) the date range (month and year) during which patients' skin samples were accessed. If patients provided informed written consent to have data from their medical records used in research, please include this information.

4. To comply with PLOS ONE submission guidelines, in your Methods section, please provide additional information regarding your statistical analyses.

For more information on PLOS ONE's expectations for statistical reporting, please see https://journals.plos.org/plosone/s/submission-guidelines.#loc-statistical-reporting.

Reviewers' comments:

Reviewer's Responses to Questions

**Comments to the Author**

1. Is the manuscript technically sound, and do the data support the conclusions?

Reviewer #1: Yes

Reviewer #2: Partly

2. Has the statistical analysis been performed appropriately and rigorously? 

Reviewer #1: Yes

Reviewer #2: No

3. Have the authors made all data underlying the findings in their manuscript fully available?

Reviewer #1: Yes

Reviewer #2: No

4. Is the manuscript presented in an intelligible fashion and written in standard English?

Reviewer #1: Yes

Reviewer #2: Yes

5. Review Comments to the Author

Reviewer #1: Summary:

The authors study genetic and epigenetic contribution to TERT gene reactivation in the context of normal skin cells and melanomas. They analyzed 44 primary tissue samples and cell-lines and quantified the methylation fraction (NGS based deep bisulphite sequencing and ddPCR assay) and mutational status (sanger sequencing and ddPCR assay) of the TERT gene. Interestingly, though the authors found that previously characterized mutations in TERT gene were unique to melanoma cell-lines, they did not seem to be correlated to expression. Indeed TERT expression seemed to be exclusive to melanoma cell-lines regardless of mutation status. Similarly, the methylation fraction of a cell-type also did not seem to be correlated to TERT expression levels. Both these findings are contrary to what has been previously characterized about TERT. In an effort to explain this discrepancy, the authors also investigated chromatin accessibility. They found that accessibility around a specific CpG was correlated to methylation and mRNA expression levels in the 8 cell-lines investigated.

Major Comments:

The study is well designed and done in both primary tissue samples and cell-lines. It furthers our understanding of the complexity of TERT gene and hence is of significance to the field.

Minor Comments:

Some things that can be done to make the manuscript more impactful:

1. The authors findings are in direct contradiction to previously carried out studies on TERT. Their accessibility experiments add another layer of complexity but do not explain the discrepancy in the relationship between methylation/mutation status and expression. It would be helpful if the authors postulate the reasons for these contradictions and discuss the impact of these findings in greater detail.

2. Figures submitted are of low resolution, as such it was hard to verify cell-line names, legends etc.

3. In Methods Statistical Analysis: mention the exact libraries used in the softwares so that the analysis can be replicated

4. Provide the raw tables that were used to make the box plots and other graphs

Reviewer #2: Salgado et al report on the association between TERT promoter mutations, DNA methylation, and chromatin accessibility in melanoma and normal cells. The authors characterize methylation fractions using bisulfite sequencing of normal skin as well as different cell types, naevi, and melanoma cell lines, finding that uveal melanoma cell lines have the highest levels of methylation (Fig. 2). Using ddPCR, they then determine that keratinocytes have a higher methylation level at a position on the TERT promoter than melanoma cell lines (Fig. 2). There was no correlation between TERT methylation at a previously characterized position and mRNA expression when comparing 34 samples and cells (Fig. 3). However, while TERT promoter mutations were identified in some melanoma cell lines and tissue (Fig. 4), there was no correlation with mRNA expression (Fig. 5). Since TERT expression was only seen in melanoma cells lines, additional analysis was conducted on these. It was then found that there was a high degree of variability in chromatin accessibility at cg11625005 between different melanoma cell lines as well as a correlation with the methylation fraction (Fig. 6). In a subset of 8 melanoma cell lines, there is a correlation between MF, chromatin accessibility, and mRNA expression of TERT. Allele specific analysis of chromatin accessibility indicated that the mutated alleles were more accessible than wildtype.

This study nicely shows that ddPCR can generate high quality methylation data that validates bisulfite sequencing. However, it is not clear that the results of this study are a big enough advance for publication. The findings are descriptive rather than mechanistic and often unsubstantiated. The allele specific data regarding chromatin accessibility are interesting but should be expanded. The correlation between high levels of chromatin accessibility at the TERT promoter and mRNA levels is also interesting but not a mechanistic finding on its own. It would be more informative to elucidate a mechanism for the varying levels of chromatin accessibility like determining transcription factor or chromatin remodeling enzyme association. Also, the nuclease assay used in this study likely determines chromatin structure at the nucleosome level and not higher order chromatin structure.

Specific points:

1. Overall, the conclusions of this study are not very clear because of the presentation of conflicting pieces of data. The lack of correlations in tissues conflicts with the correlations found in cell lines. The discussion should do a better job in explaining the physiological significance of the latter studies on a subset of melanoma cell lines.

2. It is interesting that there is allele specific differences in chromatin accessibility. Does this correlate with allele specific mRNA expression? Is there a correlation between ETS/TCF binding and chromatin accessibility at the mutated allele? It is possible that chromatin immunoprecipitations to investigate ETS binding may be informative. Otherwise, the authors should investigate an alternative hypothesis for investigating a mechanism.

3. The graphs in Fig. 6A should include error bars and both Fig.6A and Fig.6F should include p values.

4. There should be more information on the different assays used, especially the chromatin accessibility assay. Is this assay based on nuclease digestion of permeablized nuclei? If so, then it does not measure higher order chromatin structure.

5. Figure legends should include more information about the tissues and cell lines used for the specific study.

6. Lines 85 to 89 in the Results section do not describe the results in Fig. 1 very well. The authors should call attention to the finding that melanocytes are among the lowest. The difference between normal melanocytes and cutaneous melanoma cells is more than comparisons between melanomas and keratinocytes or fibroblasts.

7. The text should be modified to avoid usage of “higher order chromatin structure”.

6. PLOS authors have the option to publish the peer review history of their article (what does this mean?). If published, this will include your full peer review and any attached files.

Reviewer #1: No

Reviewer #2: No

---

## [Author Response · Author response to Decision Letter 0]

18 Feb 2020

Dear Editor, 

First we would like to express our appreciation for considering our manuscript entitled “Interplay between TERT promoter mutations and methylation culminates in chromatin accessibility and TERT expression” for publication in PLOS ONE and for the thoughtful reviews provided by the referees. We have attempted to address all comments and adapted the manuscript accordingly. Please find our responses below:

Reviewer #1: 

The authors study genetic and epigenetic contribution to TERT gene reactivation in the context of normal skin cells and melanomas. They analyzed 44 primary tissue samples and cell-lines and quantified the methylation fraction (NGS based deep bisulphite sequencing and ddPCR assay) and mutational status (sanger sequencing and ddPCR assay) of the TERT gene. Interestingly, though the authors found that previously characterized mutations in TERT gene were unique to melanoma cell-lines, they did not seem to be correlated to expression. Indeed TERT expression seemed to be exclusive to melanoma cell-lines regardless of mutation status. Similarly, the methylation fraction of a cell-type also did not seem to be correlated to TERT expression levels. Both these findings are contrary to what has been previously characterized about TERT. In an effort to explain this discrepancy, the authors also investigated chromatin accessibility. They found that accessibility around a specific CpG was correlated to methylation and mRNA expression levels in the 8 cell-lines investigated.

Major Comments:

The study is well designed and done in both primary tissue samples and cell-lines. It furthers our understanding of the complexity of TERT gene and hence is of significance to the field.

Answer:

We thank the reviewer for the positive and constructive comments.

Minor Comments:

Some things that can be done to make the manuscript more impactful:

1. The authors findings are in direct contradiction to previously carried out studies on TERT. Their accessibility experiments add another layer of complexity but do not explain the discrepancy in the relationship between methylation/mutation status and expression. It would be helpful if the authors postulate the reasons for these contradictions and discuss the impact of these findings in greater detail.

Answer:

In our study we explore a big cohort encompassing both control/benign samples and melanoma cell lines. 

Previous studies stated absence of methylation in the TERTp in benign samples, while we found high methylation fractions (MF) in some healthy skin samples, such as normal skin (~30%), naevi (~30%) and cultured keratinocytes (~50%). In fact, in keratinocytes, the MF was as high as in cutaneous melanoma cell lines. We found this result quite intriguing, but we think it can be explained by technical differences and differences in interpretation. Whereas we used a validated quantitative approach previous studies use semi-quantitative approaches in which they interpreted a β-value of ~0.4 in normal tissue as not methylated. Therefore, the measurements seem to be similar to previous studies and it is rather a difference in interpretation. Moreover, in accordance to previous studies none of the human-derived benign samples neither harbour TERTp mutation nor expressed TERT, thereby supporting the basic oncological concept that a benign cell does not undergo undefined proliferation. 

Analysis of tumour cell lines revealed a wide range of promoter methylation levels (from 5% to 100%). Besides promoter methylation, somatic mutations are also known to be correlated with TERT reactivation. Therefore, we characterized the TERTp mutational status of the sample cohort. While 12 out of 25 showed C250T mutation and 5 the C228T mutations, 8 out of 25 were WT. All melanoma cell lines showed TERT expression, (n=25) irrespective of mutation and methylation status, supporting that there are additional telomerase-activating mechanisms involved in cancer besides methylation and mutations in TERTp [1-3]. This holistic model in which mutation, methylation of TERTp and other unidentified mechanisms might explain TERT expression in WT melanoma cell lines. 

Schwartz et al. state that the degree of chromatin folding is correlated with gene transcription and is thought to impact the regulation of DNA-dependent processes [4]. Other studies demonstrated disease-dependent differences in nucleosome positions in tissues and cell lines, suggesting an important role of chromatin architecture on cell fate [5]. Moreover, chromatin accessibility is modulated locally, since nuclease-accessible sites were enriched at gene regulatory elements, such as active TSS, indicating a link between local chromatin accessibility and gene regulation [4]. TERTp is reported to coincide with a DNAse 1 hypersensitive locus as evidenced in publicly available data (UCSC database, DHS track) which supports that accessibility at this site is involved in gene regulation. 

Since the accessibility experiments require cell culturing we were not able to perform it in the benign cells since the cultured keratinocytes, fibroblasts and melanocytes are isolated at low passages. Thus, we explored the level of chromatin accessibility and TERT expression in 8 cell lines with different methylation/mutation patterns. In fact, we found a positive correlation between chromatin accessibility and methylation levels as well as mRNA expression that ultimately explains the correlation between methylation fraction and TERT expression. In WT TERT-expressing uveal melanoma cell lines, the methylation of the whole TERTp region is close to 100% with a significantly higher chromatin accessibility compared to TERTp-mutated cell lines. Our results also show that mutated alleles in heterozygous cell lines may be more accessible and allows further dissection of the underlying mechanisms that facilitate mono-allelic expression. 

Interestingly, these results suggest a complex model in which TERT expression requires either a widely open chromatin state in TERTp-WT samples due to hypermethylation throughout the promoter leading to biallelically TERT activation or mono-allelic expression of the accessible mutated allele in combination with moderate (probably allele-specific) methylation fraction. Most of these results are in agreement with the findings from Stern et al. [6]. 

This model supported our view that the integration of different aspects are definitely more important than studying each one separately. 

 To further address this comment we have clarified the assumptions underlying this study and refined the conclusions in the manuscript text. 

2. Figures submitted are of low resolution, as such it was hard to verify cell-line names, legends etc.

Answer:

 We have verified the figures and adapted the figure resolution, the font size of cell lines and legends in agreement with journal requirements.

3. In Methods Statistical Analysis: mention the exact libraries used in the softwares so that the analysis can be replicated.

Answer:

In this study we used the GraphPad Prism software (version 8.0.1 for Windows, GraphPad Software, CA, USA) to perform all the statistical tests. Prism 8 has a wide library of analysis and in our paper we have used the linear regressions and correlations, one-way ANOVA, multiple t-tests (one t-test per cell line) and two-tailed unpaired t-test. A p-value<0.05 was considered statistically significant.

The methylation fraction obtained using ddPCR was calculated with 95% confidence interval by dividing the CNV of the digested sample with that of the paired undigested sample. Raw data was uploaded in online digital PCR management and analysis application Roodcom WebAnalysis (version 1.9.4, https://www.roodcom.nl/webanalysis/). This application provides a module to calculate methylation fraction in digital PCR with MSRE [7].

 A detailed statistical analysis subsection meeting the journal requirements has been added to the material and methods section of the manuscript. 

4. Provide the raw tables that were used to make the box plots and other graphs

Answer:

 To address this point a supplementary excel file has been created containing the raw data behind each plot in different sheets. 

Reviewer #2: 

Salgado et al report on the association between TERT promoter mutations, DNA methylation, and chromatin accessibility in melanoma and normal cells. The authors characterize methylation fractions using bisulfite sequencing of normal skin as well as different cell types, naevi, and melanoma cell lines, finding that uveal melanoma cell lines have the highest levels of methylation (Fig. 2). Using ddPCR, they then determine that keratinocytes have a higher methylation level at a position on the TERT promoter than melanoma cell lines (Fig. 2). There was no correlation between TERT methylation at a previously characterized position and mRNA expression when comparing 34 samples and cells (Fig. 3). However, while TERT promoter mutations were identified in some melanoma cell lines and tissue (Fig. 4), there was no correlation with mRNA expression (Fig. 5). Since TERT expression was only seen in melanoma cells lines, additional analysis was conducted on these. It was then found that there was a high degree of variability in chromatin accessibility at cg11625005 between different melanoma cell lines as well as a correlation with the methylation fraction (Fig. 6). In a subset of 8 melanoma cell lines, there is a correlation between MF, chromatin accessibility, and mRNA expression of TERT. Allele specific analysis of chromatin accessibility indicated that the mutated alleles were more accessible than wildtype.

 study nicely shows that ddPCR can generate high quality methylation data that validates bisulfite sequencing. However, it is not clear that the results of this study are a big enough advance for publication. The findings are descriptive rather than mechanistic and often unsubstantiated. The allele specific data regarding chromatin accessibility are interesting but should be expanded. The correlation between high levels of chromatin accessibility at the TERT promoter and mRNA levels is also interesting but not a mechanistic finding on its own. It would be more informative to elucidate a mechanism for the varying levels of chromatin accessibility like determining transcription factor or chromatin remodeling enzyme association. Also, the nuclease assay used in this study likely determines chromatin structure at the nucleosome level and not higher order chromatin structure.

Specific points:

1. Overall, the conclusions of this study are not very clear because of the presentation of conflicting pieces of data. The lack of correlations in tissues conflicts with the correlations found in cell lines. The discussion should do a better job in explaining the physiological significance of the latter studies on a subset of melanoma cell lines. 

Answer:

Throughout the last 20 years many processes have been identified as having an effect in the regulation of TERT gene, namely histone modifications, such as acetylation, methylation, phosphorylation, and ubiquitinization [8], CpG methylation at TERTp and TERTp mutations. TERT expression was associated with hyperacetylation of core histones at the TERTp [9, 10]. Methylation at lysine 9 of histone H3 (H3K9) and lysine 20 of H4 (H4K20) were features of telomerase-negative immortal cells, whereas methylation at lysine 4 of histone H3 (H3K4) was usual in telomerase-positive cells [11]. Depletion of the histone methyltransferase SMYD3 leading to reduction of histone H3K4 methylation at TERTp led to downregulation of TERT [12] and inhibition of demethylase LSD1 led to increase of methylation at H3K4 and an TERT upregulation [13]. 

A CpG island covering the TERTp has caught attention of the scientific community since in cancer cells and cell lines [14, 15] it was often methylated but the correlation with transcription has been hard to make due to the CG-rich sequence. It has been suggested that hypermethylation was implicated in the positive regulation of the TERTp [16] possibly due to the hampering of binding of transcriptional repressors [17].

Only few years ago with the advent of NGS-technologies, namely NGS-based bisulfite sequencing and nowadays the emerging techniques, as ddPCR that we present in our study, it became possible to draw robust conclusions on how TERTp methylation affects TERT gene regulation [6]. 

Within the last decade with the discovery of TERTp mutations and the correlation with TERT upregulation, due to the generation of new transcription factors binding motifs, another layer of complexity arose and allowed to clarify the mechanism in some types of cancer [2, 3]. 

Zhu and colleagues postulated that the repressive chromatin environment and nucleosomal conformation appear to be one of the major mechanisms that tightly suppress the TERT gene in majority of human somatic cells [8]. 

We can see that all the aforementioned studies have contributed at multiple levels of gene regulation to reveal the complex process of regulating TERT expression. 

In our study, we were able to develop and validate innovative ddPCR-based assays to assess TERT promoter methylation and chromatin accessibility in a quantitative fashion.

While previous studies stated absence of methylation in the TERTp in benign samples despite β-values of ~0.4 in methylation array analysis, we presented significant methylation fractions (MF) in some healthy skin samples, such as normal skin, naevi and keratinocytes. However, in accordance to previous studies none of the human-derived benign samples neither harbour TERTp mutation nor expressed TERT, thereby supporting the basic oncological concept that a benign cell does not undergo undefined proliferation. 

Analysis of tumour cell lines revealed a wide range of promoter methylation levels (from 5% to 100%) and different TERTp mutational status: 12 out of 25 showed C250T mutation, 5 the C228T mutations and 8 out of 25 were WT. However, all tumour cell lines showed TERT expression, (n=25) irrespective of mutation and methylation status, supporting that there are other telomerase-activating mechanisms in cancer besides methylation and mutation [1-3].

Then, we comprehensively analysed the chromatin status in the light of current knowledge. Back in 2004, Wang and Zhu have described the chromatin environment to be critical for the tight regulation of the TERT gene, stating that in telomerase-positive cells, the TERT transcription was accompanied by the appearance of a major DNase I hypersensitive site (DHS) at the core promoter [18, 19]. 

In the present study, from the chromatin accessibility assay, we could observe an interplay between DNA methylation and presence of TERTp mutations culminates in different levels of accessibility and thus TERT expression. Our results also suggest that mutated alleles in heterozygous tumors are more accessible compared to WT allele, favoring mono-allelic expression.

These results suggest a complex model in which TERT expression requires either a widely open chromatin state in TERTp-WT samples due to hypermethylation throughout the promoter leading to biallelically TERT activation or mono-allelic expression of the accessible mutated allele in combination with moderate (probably allele-specific) methylation fraction. Most of these results are in agreement with the findings from Stern et al. [6]. 

We believe that the major interest is to integrate all the data to understand how this important gene involved in immortalization and undefined proliferation is regulated. We hereby provide a holistic model that is based on observations in tissue and cell lines. This offers the possibility to investigate TERT regulation in vitro and in vivo in much higher detail. However, it is beyond of the scope of this article to resolve the actual mechanism. 

 We have clarified these important points by adapting the discussion section. 

2. It is interesting that there is allele specific differences in chromatin accessibility. Does this correlate with allele specific mRNA expression? Is there a correlation between ETS/TCF binding and chromatin accessibility at the mutated allele? It is possible that chromatin immunoprecipitations to investigate ETS binding may be informative. Otherwise, the authors should investigate an alternative hypothesis for investigating a mechanism.

Answer:

We thank the reviewer for providing this relevant suggestion. In fact, our results suggest that there are allele specific differences in chromatin accessibility. Showing a correlation with allele specific mRNA expression would be of interest, however the informative cell lines do not provide heterogeneity in the transcribed gene and hence does not allow allele-specific quantification. Differential CTCF binding is definitely a good possibility but is beyond the scope of the present study. However, we state that further investigations are needed to fully disclose this mechanism. 

3. The graphs in Fig. 6A should include error bars and both Fig.6A and Fig.6F should include p values.

Answer:

The reviewer’s comment made us see that Figure 6 needs to be adapted. Therefore, in the figure 6A the error bars were added. The GAPDH bars were removed and mentioned in the legend. The TERT methylation region values were normalized and compared to withdrawn GAPDH data in order to generate p-values. 

Regarding the Fig 6F, error bars were added in the expression bars. Moreover, we decided to simplify the figure by showing the subtraction between mutational fraction in undigested and digested divided by undigested fraction as follows:

Accessibility of mutant allele =(mutational fraction undigested-mutational fraction digested )/(mutational fraction undigested)×100

Statistically significant differences in mutational fraction before and after nuclease digestion are indicated by asterisks.

 Based on this remark we have added an ‘allele-specific chromatin accessibility’ subsection in the material and methods section.

4. There should be more information on the different assays used, especially the chromatin accessibility assay. Is this assay based on nuclease digestion of permeabilized nuclei? If so, then it does not measure higher order chromatin structure.

Answer:

In the material and methods section we state that chromatin analysis was performed according to manufacturer’s instructions followed by a brief description of the method used “The cells were permeabilized and treated with EpiQ chromatin digestion buffer with or without nuclease for 1 hour at 37°C”. In fact, the kit used to evaluate the chromatin status is based on nuclease digestion of permeabilized nuclei. Since it is part of a commercial kit the type of nuclease is not revealed. Like that we could not explore the reported function and better interpret the readout of our experiment, as performed for MNase endonuclease used in the study of Schwartz et al., 2018 [4]. However, since the site that we investigate, overlaps with a DNase 1 hypersensitive site we assume that our test can also detect a repressive chromatin. Moreover, this is supported by a good correlation between gene accessibility and gene expression (Fig. 6C).

Therefore, through this methodology we were able to measure chromatin accessibility that is modulated on local scale, with nuclease-accessible sites enriched at gene regulatory elements, such as active TSS [4]. To accurately measure the higher order chromatin structure we would need to perform a genome-wide chromatin immunoprecipitation (ChIP) analyses to assess the dynamics of higher-order chromatin organization [20] or even more precise a DNA-labelling method to stain the higher-order chromatin clusters and fibers that enables the reconstruction of ultrastructure to be visualized in the nucleus of human cells in situ, such as the ChromEMT (ChromEM tomography) [21]. 

 As above mentioned, the material and methods section has been edited in order to give more detailed information about the assays used, namely about the chromatin accessibility assay for which an ‘allele-specific chromatin accessibility’ subsection has been added.

5. Figure legends should include more information about the tissues and cell lines used for the specific study.

Answer:

 The figure legends have been adapted and now they incorporate information about cell lines and tissues used and also some details about statistical analysis performed.

6. Lines 85 to 89 in the Results section do not describe the results in Fig. 1 very well. The authors should call attention to the finding that melanocytes are among the lowest. The difference between normal melanocytes and cutaneous melanoma cells is more than comparisons between melanomas and keratinocytes or fibroblasts.

Answer:

We thank the reviewer for this constructive comment. We have stressed the difference in methylation fraction between keratinocytes and melanoma cell lines, since has been reported that there is an absence of methylation at these location in non-malignant samples [22, 23], while we found similar methylation levels in both keratinocytes and melanoma cell lines. However, we agree that since melanocytes are the cellular origin of melanoma it is quite remarkable that we found this striking difference in methylation between these two types of samples. 

 The explanation of Fig. 1 in the results section has been extended.

7. The text should be modified to avoid usage of “higher order chromatin structure”.

Answer:

We thank the reviewer for drawing attention to this potentially confusing point. Most textbooks present the long-lasting model in which the primary DNA-nucleosome subunits progressively fold into discrete higher-order chromatin fibers and, ultimately, mitotic chromosomes. Despite all the research carried out the chromatin organization is still an enigma. However, recent literature reports that no global differences in DNA packaging do exist, challenging the model of hierarchically organized higher-order structures of chromatin [4, 21]. 

 The manuscript text has been adapted in order to avoid the expression. 

We thank the reviewers for their thoughtful comments. Addressing the questions and comments has helped us to clarify parts of the study. We have the impression that the adaptations have improved the manuscript considerably. 

Kind regards,

Catarina Salgado (on behalf of all authors) 

 

References:

1. Lee DD, Leao R, Komosa M, Gallo M, Zhang CH, Lipman T, et al. DNA hypermethylation within TERT promoter upregulates TERT expression in cancer. The Journal of clinical investigation. 2019;129(4):1801.

2. Horn S, Figl A, Rachakonda PS, Fischer C, Sucker A, Gast A, et al. TERT promoter mutations in familial and sporadic melanoma. Science (New York, NY). 2013;339(6122):959-61.

3. Huang FW, Hodis E, Xu MJ, Kryukov GV, Chin L, Garraway LA. Highly recurrent TERT promoter mutations in human melanoma. Science (New York, NY). 2013;339(6122):957-9.

4. Schwartz U, Nemeth A, Diermeier S, Exler JH, Hansch S, Maldonado R, et al. Characterizing the nuclease accessibility of DNA in human cells to map higher order structures of chromatin. Nucleic acids research. 2019;47(3):1239-54.

5. Valouev A, Johnson SM, Boyd SD, Smith CL, Fire AZ, Sidow A. Determinants of nucleosome organization in primary human cells. Nature. 2011;474(7352):516-20.

6. Stern JL, Paucek RD, Huang FW, Ghandi M, Nwumeh R, Costello JC, et al. Allele-Specific DNA Methylation and Its Interplay with Repressive Histone Marks at Promoter-Mutant TERT Genes. Cell reports. 2017;21(13):3700-7.

7. Nell RJ, Steenderen Dv, Menger NV, Weitering TJ, Versluis M, van der Velden PA. Quantification of DNA methylation using methylation-sensitive restriction enzymes and multiplex digital PCR. 2019:816744.

8. Zhu J, Zhao Y, Wang S. Chromatin and epigenetic regulation of the telomerase reverse transcriptase gene. Protein & cell. 2010;1(1):22-32.

9. Xu D, Popov N, Hou M, Wang Q, Bjorkholm M, Gruber A, et al. Switch from Myc/Max to Mad1/Max binding and decrease in histone acetylation at the telomerase reverse transcriptase promoter during differentiation of HL60 cells. Proceedings of the National Academy of Sciences of the United States of America. 2001;98(7):3826-31.

10. Wang S, Hu C, Zhu J. Transcriptional silencing of a novel hTERT reporter locus during in vitro differentiation of mouse embryonic stem cells. Molecular biology of the cell. 2007;18(2):669-77.

11. Atkinson SP, Hoare SF, Glasspool RM, Keith WN. Lack of telomerase gene expression in alternative lengthening of telomere cells is associated with chromatin remodeling of the hTR and hTERT gene promoters. Cancer research. 2005;65(17):7585-90.

12. Liu C, Fang X, Ge Z, Jalink M, Kyo S, Bjorkholm M, et al. The telomerase reverse transcriptase (hTERT) gene is a direct target of the histone methyltransferase SMYD3. Cancer research. 2007;67(6):2626-31.

13. Zhu Q, Liu C, Ge Z, Fang X, Zhang X, Straat K, et al. Lysine-specific demethylase 1 (LSD1) Is required for the transcriptional repression of the telomerase reverse transcriptase (hTERT) gene. PloS one. 2008;3(1):e1446.

14. Devereux TR, Horikawa I, Anna CH, Annab LA, Afshari CA, Barrett JC. DNA methylation analysis of the promoter region of the human telomerase reverse transcriptase (hTERT) gene. Cancer research. 1999;59(24):6087-90.

15. Dessain SK, Yu H, Reddel RR, Beijersbergen RL, Weinberg RA. Methylation of the human telomerase gene CpG island. Cancer research. 2000;60(3):537-41.

16. Guilleret I, Yan P, Grange F, Braunschweig R, Bosman FT, Benhattar J. Hypermethylation of the human telomerase catalytic subunit (hTERT) gene correlates with telomerase activity. International journal of cancer. 2002;101(4):335-41.

17. Renaud S, Loukinov D, Bosman FT, Lobanenkov V, Benhattar J. CTCF binds the proximal exonic region of hTERT and inhibits its transcription. Nucleic acids research. 2005;33(21):6850-60.

18. Wang S, Zhu J. Evidence for a relief of repression mechanism for activation of the human telomerase reverse transcriptase promoter. The Journal of biological chemistry. 2003;278(21):18842-50.

19. Wang S, Zhu J. The hTERT gene is embedded in a nuclease-resistant chromatin domain. The Journal of biological chemistry. 2004;279(53):55401-10.

20. Woodcock CL, Ghosh RP. Chromatin higher-order structure and dynamics. Cold Spring Harbor perspectives in biology. 2010;2(5):a000596.

21. Ou HD, Phan S, Deerinck TJ, Thor A, Ellisman MH, O'Shea CC. ChromEMT: Visualizing 3D chromatin structure and compaction in interphase and mitotic cells. Science (New York, NY). 2017;357(6349).

22. Castelo-Branco P, Choufani S, Mack S, Gallagher D, Zhang C, Lipman T, et al. Methylation of the TERT promoter and risk stratification of childhood brain tumours: an integrative genomic and molecular study. The Lancet Oncology. 2013;14(6):534-42.

23. Barthel FP, Wei W, Tang M, Martinez-Ledesma E, Hu X, Amin SB, et al. Systematic analysis of telomere length and somatic alterations in 31 cancer types. Nature genetics. 2017;49(3):349-57.

---

## [Decision Letter · Decision Letter 1]

24 Mar 2020

Interplay between TERT promoter mutations and methylation culminates in chromatin accessibility and TERT expression

PONE-D-19-33667R1

Dear Dr. Salgado,

We are pleased to inform you that your manuscript has been judged scientifically suitable for publication and will be formally accepted for publication once it complies with all outstanding technical requirements.

With kind regards,

Srinivas Saladi, Ph.D.

Academic Editor

PLOS ONE

Additional Editor Comments (optional):

Reviewers' comments:

Reviewer's Responses to Questions

**Comments to the Author**

1. If the authors have adequately addressed your comments raised in a previous round of review and you feel that this manuscript is now acceptable for publication, you may indicate that here to bypass the “Comments to the Author” section, enter your conflict of interest statement in the “Confidential to Editor” section, and submit your "Accept" recommendation.

Reviewer #1: All comments have been addressed

Reviewer #2: All comments have been addressed

2. Is the manuscript technically sound, and do the data support the conclusions?

Reviewer #1: Yes

Reviewer #2: (No Response)

3. Has the statistical analysis been performed appropriately and rigorously? 

Reviewer #1: Yes

Reviewer #2: (No Response)

4. Have the authors made all data underlying the findings in their manuscript fully available?

Reviewer #1: Yes

Reviewer #2: (No Response)

5. Is the manuscript presented in an intelligible fashion and written in standard English?

Reviewer #1: Yes

Reviewer #2: (No Response)

6. Review Comments to the Author

Reviewer #1: The authors have done a good job of addressing all the comments. They have explained the discrepancy between their results and previously published studies. They have also updated the methods section to be more comprehensive about the biochemical experiments and statistical tests performed.

Reviewer #2: (No Response)

7. PLOS authors have the option to publish the peer review history of their article (what does this mean?). If published, this will include your full peer review and any attached files.

Reviewer #1: No

Reviewer #2: No

---

## [Editor Report · Acceptance letter]

25 Mar 2020

PONE-D-19-33667R1 

Interplay between *TERT* promoter mutations and methylation culminates in chromatin accessibility and *TERT* expression 

Dear Dr. Salgado:

I am pleased to inform you that your manuscript has been deemed suitable for publication in PLOS ONE. Congratulations! Your manuscript is now with our production department. 

With kind regards,

on behalf of

Dr. Srinivas Saladi 

Academic Editor

PLOS ONE